# Impact of different blood pressure targets on cerebral hemodynamics in septic shock: A prospective pilot study protocol—SEPSIS-BRAIN

**Pedro Cury**[1,2,3]☉*, **Rogério da Hora Passos**[4], **Fernanda Alves**[1,2], **Sérgio Brasil**[5], **Gustavo Frigieri**[6], **Fabio S. Taccone**[7], **Ronney B. Panerai**[8,9], **Juliana Caldas**[1,2,3]☉*

**1** Critical Care Unit, D'Or Institute for Research and Education (IDOR), Salvador, Brazil, **2** Hospital São Rafael, Salvador, Brazil, **3** Bahiana—School of Medicine and Public Health, Salvador, Brazil, **4** Hospital Israelita Albert Einstein, São Paulo, SP, Brazil, **5** University of Sao Paulo, Sao Paulo, Brazil, **6** Medical Investigation Laboratory 62, School of Medicine, University of São Paulo, São Paulo, Brazil, **7** Department of Intensive Care, Hospital Erasme, Brussels, Belgium, **8** Department of Cardiovascular Sciences, University of Leicester, Leicester, United Kingdom, **9** NIHR Leicester Biomedical Research Centre, Glenfield Hospital, Leicester, United Kingdom

☉ These authors contributed equally to this work.
* caldas.juliana@gmail.com (JC); pedrovcury@gmail.com (PC)

**Data Availability Statement:** All relevant data are within the paper and its Supporting Information files.

## Abstract

### Introduction

Septic shock, a life-threatening condition, can result in cerebral dysfunction and heightened mortality rates. In these patients, disturbances in cerebral hemodynamics, as reflected by impairment of myogenic cerebral autoregulation (CA), metabolic regulation, expressed by critical closing pressure (CrCP) and reductions in intracranial compliance (ICC), can adversely impact septic shock outcomes. The general recommendation is to maintain a target mean arterial pressure (MAP) of 65 mmHg but the effect of different MAP targets on cerebral hemodynamics in these patients is not clear and optimal targets might be dependent on the status of CA. This protocol aims to assess the cerebral hemodynamics profile at different pressure targets in septic shock patients.

### Methods

Prospective, non-randomized, single-center trial, which will study cerebral hemodynamics in patients with septic shock within 48 hours of its onset. Patients will be studied at their baseline MAP and at three MAP targets (T1: 65, T2: 75, T3: 85 mmHg). Cerebral hemodynamics will be assessed by transcranial Doppler (TCD) and a skull micro-deformation sensor (B4C). Dynamic CA will be expressed by the autoregulation index (ARI), calculated by transfer function analysis, using fluctuations of MAP as input and corresponding oscillations in cerebral blood velocity (CBv). The instantaneous relationship between arterial blood pressure and CBv will be used to estimate CrCP and resistance-area product (RAP) for each cardiac cycle using the first harmonic method. The B4C will access ICC by intracranial pressure waveforms (P2/P1). The primary aim is to assess cerebral hemodynamics (ARI, CrCP,

**Funding:** The authors received no specific funding for this work.

**Competing interests:** The authors have declared that no competing interests exist.

**Abbreviations:** ARI, Autoregulation index; B4C, Brain4care; CA, Cerebral autoregulation; CBv, Cerebral blood velocity; CrCP, Critical closing pressure; dCA, dynamic Cerebral autoregulation; ICU, Intensive care unit; ICC, Intracranial compliance; IQRs, Interquartile ranges; MAP, Mean arterial pressure; RAP, Resistance-area product; SAE, Serious adverse event; TCD, Transcranial Doppler.

RAP, and P2/P1) at different targets of MAP in septic shock patients. Our secondary objective is to assess cerebral hemodynamics at 65mmHg (target recommended by guidelines). In addition, we will assess the correlation between markers of organ dysfunction (such as lactate levels, vasoactive drugs usage, SOFA score, and delirium) and CA.

## Ethics and dissemination

The results of this study may help to understand the effect of the recommended MAP and variations in blood pressure in patients with septic shock and impaired CA and ICC. Furthermore, the results can assist large trials in establishing new hypotheses about neurological management in this group of patients. Approval was obtained from the local Ethics Committee (28134720.1.0000.0048). It is anticipated that the results of this study will be presented at national and international conferences and will be published in peer-reviewed journals in 2024 and 2025.

## Trial registration

**Trial registration number:** NCT05833607. https://clinicaltrials.gov/study/NCT05833607.

## Introduction

Cerebral dysfunction in sepsis, spanning delirium to coma, is closely linked to elevated mortality rates [1,2]. The underlying pathophysiological alterations contributing to patient deterioration and death are complex and multifaceted. Nevertheless, several studies have highlighted an association between altered cerebral hemodynamics and the onset of sepsis and shock [3–5].

Recent findings indicate that patients with septic shock exhibit compromised dynamic cerebral autoregulation (dCA) [3,4]. A reduced autoregulation index (ARI)relative to healthy individuals also showed an inverse correlation between the Sequential Organ Failure Assessment (SOFA) score Another observational trial with 100 patients using the mean velocity index also demonstrated an association between brain dysfunction, sepsis, and impairment of dCA [4].

DCA characterizes the brain's ability to maintain consistent cerebral blood flow (CBF), notwithstanding variations in mean arterial pressure (MAP) [6]. This capability acts as a protective mechanism, shielding the brain tissue from extremes of CBF changes and the associated risks of both hyperperfusion and hypoperfusion due to MAP fluctuations [7].

Patients with sepsis are susceptible not only to dCA impairments but also to elevated intracranial pressure (ICP) and brain edema [8]. Such pathophysiological shifts result in decreased intracranial compliance (ICC), exacerbating the already compromised dCA. Deterioration of cerebral perfusion can also be reflected by changes in critical closing pressure (CrCP), which represents the arterial blood pressure level at which CBF reaches zero [9], and resistance-area product (RAP), which represents the slope of the instantaneous pressure–velocity relationship for each cardiac cycle [10]. Alterations in CrCP can reflect disturbances in metabolic CBF control and have serious deleterious effects on cerebral perfusion by leading to hyper or hypoperfusion [10,11].

While current guidelines from the Surviving Sepsis Campaign advocate for a target MAP of 65 mmHg to secure tissue perfusion [12], the empirical basis for this MAP threshold is scant. This is primarily due to the limited research into cerebral hemodynamic dysfunction in septic patients, highlighting a significant gap in our understanding. Some trials have sought to

establish an association between systemic and cerebral factors. Masse et al. did not find different measures of CBF in septic patients when comparing those who used vasoactive drugs to those who did not, indicating that MAP is not the sole contributing factor in ensuring CBF [13]. Another study, in which dobutamine was infused during transcranial Doppler (TCD) monitoring, suggests that dobutamine increases CBF but not cerebral oxygen consumption (estimated by jugular bulb oximetry) in stable septic patients [14]. It is important to highlight that these studies did not evaluate either the dCA or ICC of their patients.

Considering the profile of critically ill patients, the use of multimodal monitoring has been increasingly recommended, especially with non-invasive tools, leading to new diagnostic hypotheses and interventions [14]. A validated non-invasive mechanical sensor (B4C) has been shown to effectively capture crucial aspects of the intracranial pressure waveform [15,16]. In a recent study, the B4C sensor was employed to predict intracranial pressure elevations with a high degree of accuracy, demonstrating comparable results to those obtained from invasive monitoring techniques [17]. Therefore, this tool has considerable potential for the management of critically ill patients undergoing multimodal, non-invasive monitoring.

In this pilot study, we aim to investigate the impact of varying MAP targets on cerebral hemodynamic parameters, utilizing non-invasive methodologies such as TCD and the B4C sensor.

## Objectives

Our primary endpoint is to evaluate the effect of various mean arterial pressure (MAP) targets on cerebral hemodynamics (ARI, CrCP, RAP, and P2/P1) in patients with septic shock. Our secondary endpoint is to assess cerebral hemodynamics at 65 mmHg and its correlation with indicators of clinical severity and neurological dysfunction.

We hypothesize that these results will present a different pattern in patients with impaired dCA and altered ICC and consider whether these parameters can be modified by different MAP levels, suggesting a personalized approach to cerebral hemodynamics.

## Trial design

We will perform a single-center prospective study with consecutive inclusion of patients with septic shock within 48 hours of its onset.

## Methods: Participants, interventions, and outcomes

### Study setting

This single-center clinical trial, a non-randomized study, will be conducted in three intensive care units (ICUs) at Hospital São Rafael, Brazil. A screening by a multidisciplinary team will be carried out daily to identify candidates successively for inclusion, and then the inclusion and exclusion criteria will be applied.

Data collection is expected to take one year and to include at least 40 patients. Measurements will be performed using Transcranial Doppler (TCD; DWL Doppler box, Germany, Serial number: DB-1964) with a 2 MHz probe, B4C (Brain4care Corp., São Carlos, Brazil), and capturing data from the multiparametric monitor (Fig 1). Clinical and laboratory data will be collected from all subjects.

The clinical trial study protocol was approved by the local Ethics Committee (28134720.1.0000.0048) and in Clinicatrials.gov (NCT05833607).

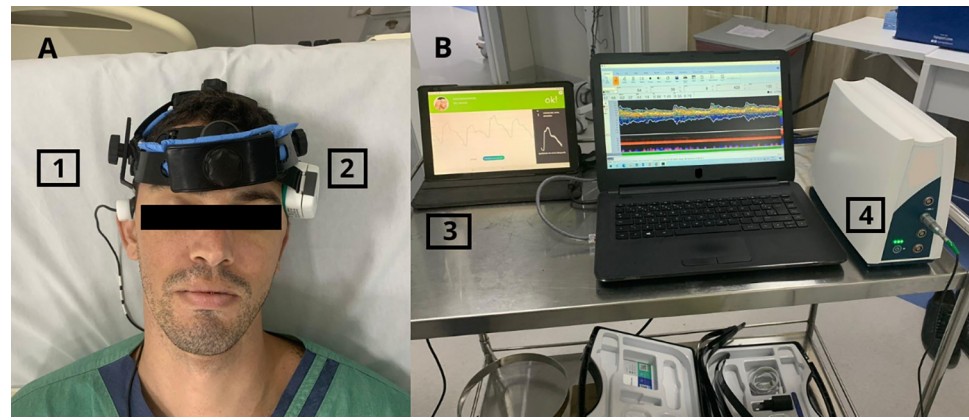

**Fig 1. Cerebral monitoring for data collection of the study.** 1) Transcranial Doppler probe maintained in position with a probe-holder; 2) Brain4care sensor; 3) Brain4care main display showing the cerebral compliance curve; 4) Transcranial Doppler device main display showing cerebral blood velocity of the middle cerebral artery. (Under a CC BY license, with permission from Dr. Thiago Passos. Vincenzo Lionetti, original copyright 2023).

**Selection criteria.** Study enrollment will include patients experiencing septic shock for up to 48 hours and still necessitating vasopressor support. Detailed inclusion and exclusion criteria are provided in Table 1.

**Who will take informed consent?.** Local investigators will present the study details to either the patient or their legal representatives, such as next of kin or legal guardians, covering all crucial trial aspects. This will ensure that a comprehensive and informed discussion takes place, facilitating a clear understanding of the study's scope and procedures.

Table 1. Selection criteria.

| Inclusion criteria | Exclusion criteria |
|---|---|
| • Age > 18 | • No temporal window for TCD |
| • Septic shock < 48h<br>• The timing will be counted from ICU admission or the start of noradrenaline infusion for patients already in critical care units. | • Hepatic or uremic encephalopathy |
| | • Pregnancy |
| | • Acute or prior neurological insult* |
| | • Exogenous intoxication |
| | • Dementia** |
| | • Severe hypercapnia***<br>• Instability for acute arrhythmia |
| | • Extracorporeal support device by modifying the pulse waveform**** |
| | • Extreme severity, with imminent risk of death |

\* Ischemic or hemorrhagic stroke, aneurysm, arteriovenous malformation, hydrocephalus, neurological surgery, central nervous system infection.

\*\* Reported or written in medical records.

\*\*\* Partial pressure of arterial carbon dioxide (PaCO2) > 65mmHg.

\*\*\*\* Intra-aortic balloon pump (IABP) and Extracorporeal membrane oxygenation (ECMO).

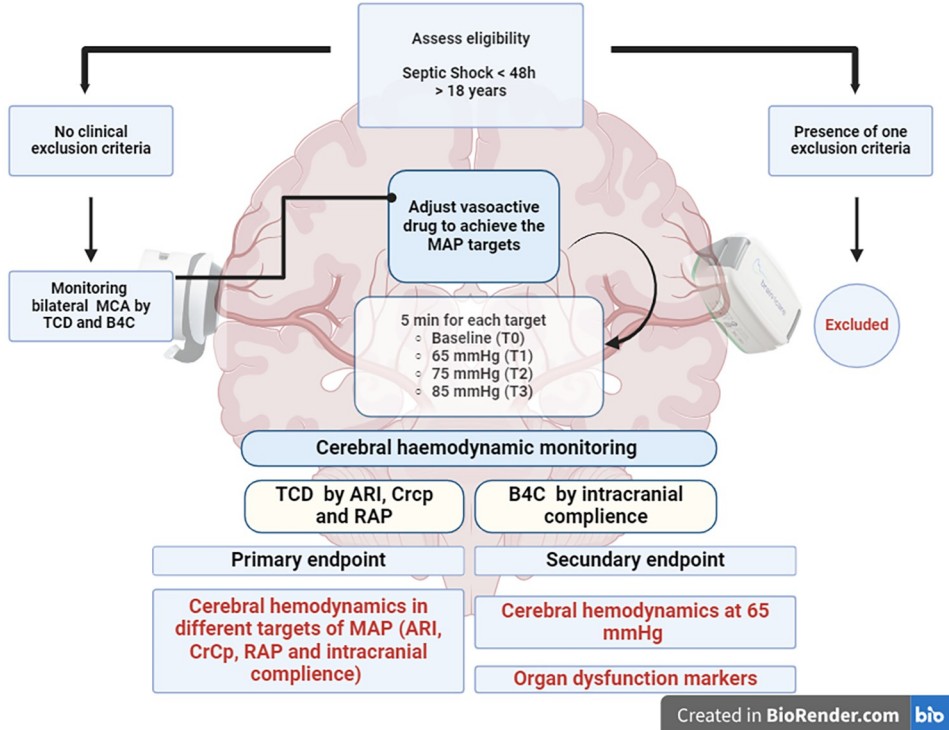

**Fig 2. Flowchart:** MCA, middle cerebral artery; TCD, transcranial Doppler; B4C, brain4care; ARI, autoregulation index; CA, cerebral autoregulation; Crcp, critical closing pressure; RAP, resistance-area product.

## Study outcome

The primary endpoint is to assess the impact of different MAP targets on cerebral hemodynamics (ARI, CrCP, RAP by TCD, and P2/P1 by B4C) in patients with septic shock.

Secondary objectives: (1) Evaluate the effects of targeting a MAP of 65 mmHg on cerebral hemodynamics in patients with septic shock. (2) Evaluate the correlation between ARI and B4C. (3) Correlate criteria of clinical severity and neurological dysfunction with cerebral hemodynamics (This will involve assessing the correlation between markers of organ dysfunction such as lactate levels, usage of vasoactive drugs, Sequential Organ Failure Assessment (SOFA) score, and delirium with CA). (4) Assess 28-day mortality.

## Participant timeline

The participant timeline is reported in Fig 2.

## Sample size

This pilot project is designed to elucidate the physiological changes correlated with different blood pressure targets within cerebral hemodynamics. The insights gained will facilitate the determination of sample sizes for subsequent studies following this protocol.

## Recruitment

The recruitment phase is set from July 2023 to June 2025. Table 2 details the project's steps. Septic shock patients starting on noradrenaline, whether newly admitted to or already in the

**Table 2. Protocol chronology.**

| | | | STUDY PERIOD | | | | |
| --- | --- | --- | --- | --- | --- | --- | --- |
| | Recruitment | Allocation | Intervention protocol (5 min on each MAP target) | | | Clinical outcome | |
| | *07/23–06/25* | *< 48 hours ICU admission* | *MAP 65 mmHg* | *MAP 75 mmHg* | *MAP 85 mmHg* | *7th day* | *28th day* |
| **ENROLMENT:** | | | | | | | |
| **Eligibility screen** | x | | | | | | |
| **Informed consent** | x | | | | | | |
| **Allocation** | | x | | | | | |
| | | | | | | | |
| **INTERVENTIONS:** | | | | | | | |
| *Induce different MAP targets* | | | x | x | x | | |
| *Monitor bilateral MCA by TCD and B4C* | | | x | x | x | | |
| | | | | | | | |
| *ASSESSMENTS* | | | | | | | |
| **Sex, age, time, time since admission to the ICU, infection site, comorbidities, APACHE, SOFA, vasopressor dosage, blood analysis** | | x | | | | | |
| **TCD: ARI, CrCP, RAP** | | | X | X | X | | |
| **B4C: P2/P1 ratio** | | | X | X | X | | |
| **Brain dysfunction** | | X | | | | x | x |
| **Ventilation free days** | | | | | | | x |

MAP: Mean arterial pressure, MCA: Middle cerebral artery, TCD: Transcranial doppler, B4C: Brain4care, ICU: Intensive care unit, APACHE: Acute Physiologic and Chronic Health Evaluation, SOFA: Sequential Organ Failure Assessment, ARI: Autoregulation index CrCP: Critical closing pressure, RAP: Resistance-area product.

ICU, are potential candidates. However, if an alternative diagnosis emerges within 48 hours, they will be removed from the candidate pool. The multiprofessional team will daily inform the research group about new vasopressor requirements, ensuring rigorous monitoring.

The locals researchers will be designated for daily data collection and interactions with patients or their families after the, ensuring that recruitment is possible within 48 hours from the onset of vasopressor therapy to optimize data gathering.

## Methods: Assignment of intervention

### Sequence generation

All patients using vasoactive drugs will be considered for inclusion in the clinical trial, subject to the application of inclusion and exclusion criteria (Table 2).

### Concealment mechanism

Given the population to be studied, the absence of a control group or the use of placebo, means that concealment is not appropriate for the study.

### Assignment of interventions: Who will be blinded?

This study will analyze the parameters of cerebral hemodynamics after data collection. Therefore, during data acquisition, it is necessary to assess the quality of hemodynamic recordings. However, the final data, involving derived parameters, will not be obtained simultaneously. Patients will be identified by numbers in the database, ensuring that the analysis remains uninfluenced by the researcher. Consequently, no additional blinding will be implemented.

## Methods: Data collection and management

Data collection on admission will include the following: demographic characteristics, comorbidities, source of infection, Sequential Organ Failure Assessment at ICU admission, temperature, hemogram lactate, hepatic and renal markers, Glasgow Coma scale or Richmond Agitation Sedation Scale, Confusion Assessment Method for the ICU scale, drug for sedation, neuromuscular blocking agents, mechanical ventilation, quantify vasoactive drug dose on collection day, arterial blood gasses before and after the protocol, presence of acute cardiac dysfunction by echocardiography and left ventricle ejection fraction. We will describe the utilization of antipsychotics or sleep-improving medications before ICU admission.

At the time of data collection, the presence of brain dysfunction associated with sepsis will be assessed by: a Glasgow scale score < 15 or when disorientation, disorganized thinking, or agitation are reported by the attending physician, regardless of sedative/analgesic use and in the absence of neurological disease (i.e., dementia, cerebrovascular disease, brain tumors, previous traumatic brain injury). Data from medical records will be used to access such information, the description of the change in the level of consciousness or the use of antipsychotics or dexmedetomidine will be interpreted as a brain dysfunction.

Systemic physiological monitoring will be performed by an arterial line, cardioscopy, pulse oximetry, and capnography (if unavailable at the time of collection, this will be described). Before starting the monitoring, the proper functioning of the arterial line will be confirmed and the site described.

Each patient will adhere to a uniform protocol where, after establishing baseline measures, the mean arterial pressure (MAP) targets will be sequentially increased in the order of 65, 75, and then 85 mmHg. This structured approach ensures sample homogeneity.

CA assessment by TCD will use a 2 MHz probe, supported by a helmet frame, the middle cerebral artery will be located through the window of the temporal bone, then it will be confirmed by depth and waveform, and one of the sides will be chosen and will be kept during the protocol. All hemodynamic parameters will be collected simultaneously and transmitted to the Doppler Box through a cable connected to the multiparameter monitor. A B4C sensor will be placed on the contralateral temporal window, which consists of a monitor that quantifies local cranial bone deformations using strain-gauge sensors. The system will be positioned on the frontotemporal region, approximately 3 cm over the first third of the orbitomeatal line, and the software will indicate the wave quality generated. The measurement of ICC will be done using the curve generated by B4C, and P2/P1 > 1 will be considered altered. The quality of measurements will be closely monitored during the duration of recordings to ascertain their acceptability for subsequent stages of off-line analysis.

The instantaneous relationship between arterial blood pressure and CBv will be used to estimate CrCP and RAP for each cardiac cycle using the first harmonic method [18]. Beat-to-beat data will be spline interpolated and resampled at 5 Hz to produce signals with a uniform time base.

Dynamic CA will be assessed using transfer function analysis, using spontaneous fluctuations of MAP as input and corresponding changes in CBv as output in recordings for each MAP target (lasting 5 min), as described previously [19,20]. The Welch method will be adopted for smoothing spectral estimates obtained with the fast Fourier transform (102.4 s segments, 50% superposition) leading to frequency-dependent estimates of coherence, gain and phase. Using the inverse fast Fourier transform, the CBv response to a step change in MAP will also be derived and compared with ten template curves proposed by Tiecks et al. [21] to estimate the ARI [22,23] Baseline cerebral hemodynamic parameters will be reported as the average over a 5-min recording at rest. Values of ARI < 4 will characterize impaired dCA.

The protocol will be interrupted if at any time during the data collection any of the adverse effects, described below, are identified.

## Plans to promote participant retention and complete follow-up

Data will be collected in a single moment, except for ventilator-free days on the 28th day, and brain dysfunction on the 7th and 28th days, which will be evaluated in the review of medical records.

In the event of no informed consent, the patient will not be included in the study.

## Data management

Each patient will be registered in the database at a dedicated website (https://www.project-redcap.org) by the research team. Regular screening of the website for missing and incorrect data will be conducted by the project leader. All collected data will be accessible via the Zenodo repository (https://zenodo.org).

Additionally, all data collected will be included in the collection form.

## Ethics statement

This project was approved by the ethics and research committee of Hospital São Rafael, Salvador—Bahia, Brazil, with registration number 28134720.1.0000.0048.

Furthermore, the individual depicted in Fig 1 has provided written informed consent (as outlined in the PLOS consent form) to publish their image alongside the manuscript.

## Confidentiality

All patients will be identified solely by a unique number assigned to them. Trial personnel are strictly prohibited from disclosing patients' names outside the local hospital or in any study-related documents. Patients will never be identified by their names; only their assigned numbers will be used for identification purposes. The subject identification code will be securely safeguarded by the principal investigator.

## Statistical methods: Outcomes

Data will be assessed for normality using the Kolmogorov–Smirnov test. Categorical variables will be compared using the Fisher exact test or chi-square test, as appropriate, and the Student t-test or the Mann–Whitney U-test will be used to compare continuous variables, as appropriate The results will be expressed as the average and standard deviation for normally distributed data, or as the median and interquartile ranges (IQRs) for non-normally distributed data.

The primary outcome will be evaluated at T0 (basal), T1 (65 mmHg), T2 (75 mmHg), and T3 (85 mmHg) using a two-way repeated-measures ANOVA. Due to the longitudinal study design and the dependence between repeated measures, mixed-effects regression models will be considered to develop a model with different pressure targets. Each component of cerebral hemodynamics (ARI, RAP, CrCP, and P1/P2) will be assessed individually. At the 65 mmHg MAP target, cerebral hemodynamics will be evaluated by mean and IQRs.

All secondary outcomes will be exploratory and analyzed using independent samples t-tests, and chi-square test, as appropriate, without additional adjustment. Linear regression, Pearson's correlation coefficient or Sperman coefficient will be used to test for association between cerebral parameters and organ dysfunction. Statistical analyses will be performed using the last version of SPSS for Windows (Chicago, USA). A p-value<0.05 will be considered statistically significant.

## Statistical methods: Analysis to handle protocol non-adherence and any statistical methods to handle missing data

After the initial data collection, mechanical ventilator-free days on the 28th day, and brain dysfunction will be recorded. Patients will be included if there is missing data. In the event of missing data, imputation will be performed by calculating the mean of the available data. Patients will be excluded from the study if the quality of the TCD recording is deemed unsatisfactory.

## Interim analysis

The project is a pilot trial and interim analysis is not planned. The adverse effects will be reported and communicated to the Ethics Committee.

# Methods: Oversight and monitoring

## Composition of the coordinating center

The principal investigator will bear the responsibility for the overall management of the study. Additionally, the study executive committee, comprising the principal investigator and project coordinator, who is affiliated with Hospital São Rafael (Brazil), will oversee the study's execution.

**Composition of the data safety monitoring committee, its role, and reporting structure.** This study will undergo monitoring by the local Ethics Committee. Any adverse effects deemed frequent or situations that may cause discomfort to the patient or their representatives will be promptly reported to the Ethics Committee for further evaluation.

**Adverse event reporting and harms.** Serious adverse events (SAEs) encompass various potential outcomes, including arrhythmia, intestinal ischemia, and acute extremity ischemia. These events tend to occur more frequently when administering exceedingly high doses of vasopressors, particularly among patients with an imminent prognosis of death within a few hours or those already exhibiting arrhythmias. Consequently, such cases are excluded from the study. Given the anticipated higher mortality rates within this patient cohort, instances of death are not reported as SAEs in this context.

Furthermore, each type of SAE and the total number of SAEs will be reported in the RED-Cap database and do not have to be reported separately.

**Frequency and plans for auditing trial conduct.** Monitoring will be conducted following national requirements to ensure the protection of patients' rights and safety, as well as compliance with the protocol, proper data collection, and accurate outcome reporting. The data monitoring process will encompass various aspects, including the inclusion rate, adherence to informed consent procedures, completeness of trial and investigators' main documents and files, recording of endpoints and SAE reports, and verification of source data such as hospital medical records, medical notes, laboratory findings, and electronic data.

**Plans for communicating important protocol amendments to relevant parties.** Relevant protocol modifications will be communicated to Ethics Committees and ClinicalTrials. org by the principal investigator via email.

**Research ethics approval.** The local Ethics Committee of São Rafael Hospital approved this multicenter study on the 18th of February 2020 with CAE number: 28134720.1.0000.0048. It was ensured that written informed consent to participate will be obtained from all participants. No deviation from the protocol has been implemented without the prior review and approval of the Ethics Committee.

**Informed consent.** Written informed consent is required from either the patient or their legal representatives before initiating the protocol. It is recommended that written informed

consent be obtained as soon as possible after ICU admission if sepsis is suspected so that inclusion can be expedited when the selection criteria are met. This procedure anticipates situations where the patient loses the ability to consent, or their caregiver is absent from the hospital.

Patients or their next of kin can withdraw consent at any time during the study without providing a specific reason. Withdrawal should not influence the standard of care. Patients who withdraw from the study are asked to provide consent for the inclusion of data collected before withdrawal.

**Additional consent provisions for the collection and use of participant data and biological specimens.** No biological specimens will be collected.

**Access to data.** All collected data will be accessible via the Zenodo repository (https://zenodo.org).

**Dissemination policy: Trial results.** After the outcome assessment collection, data will be analyzed within 3 months and presented to all investigators before publication. Confidentiality of the data will be maintained, and we guarantee that for 5 years, they will be made available upon request. Important steps will involve disseminating the obtained results among healthcare professionals, patients, and policymakers to contribute to the enhancement of clinical practice in this setting.

## Supporting information

**S1 Fig. Cerebral monitoring for data collection of the study.** Cerebral monitoring for data collection of the study. 1) Transcranial Doppler probe maintained in position with a probe-holder; 2) Brain4care sensor; 3) Brain4care main display showing the cerebral compliance curve; 4) Transcranial Doppler device main display showing cerebral blood velocity of the middle cerebral artery. (Under a CC BY license, with permission from Dr. Thiago Passos. Vincenzo Lionetti, original copyright 2023).
(PDF)

**S2 Fig. The participant timeline.** Flowchart: MCA, middle cerebral artery; TCD, transcranial Doppler; B4C, brain4care; ARI, autoregulation index; CA, cerebral autoregulation; Crcp, critical closing pressure; RAP, resistance-area product.
(PDF)

**S1 Table. Protocol Chronology.** MAP: mean arterial pressure, MCA: middle cerebral artery, TCD: transcranial doppler, B4C: brain4care, ICU: intensive care unit, APACHE: Acute Physiologic and Chronic Health Evaluation, SOFA: Sequential Organ Failure Assessment, ARI: autoregulation index CrCP: critical closing pressure, RAP: resistance-area product.
(PDF)

**S2 Table. Selection criteria.** * Ischemic or hemorrhagic stroke, aneurysm, arteriovenous malformation, hydrocephalus, neurological surgery, central nervous system infection. ** Reported or written in medical records. *** Partial pressure of arterial carbon dioxide ($PaCO_2$) > 65mmHg. **** Intra-aortic balloon pump (IABP) and Extracorporeal membrane oxygenation (ECMO).
(PDF)

## Author Contributions

**Conceptualization:** Rogério da Hora Passos, Sérgio Brasil, Gustavo Frigieri, Ronney B. Panerai.

**Data curation:** Pedro Cury.

**Formal analysis:** Pedro Cury, Fernanda Alves, Ronney B. Panerai, Juliana Caldas.

**Investigation:** Pedro Cury, Fernanda Alves, Juliana Caldas.

**Methodology:** Sérgio Brasil, Juliana Caldas.

**Project administration:** Pedro Cury, Rogério da Hora Passos, Fernanda Alves, Ronney B. Panerai, Juliana Caldas.

**Resources:** Rogério da Hora Passos.

**Supervision:** Sérgio Brasil, Ronney B. Panerai, Juliana Caldas.

**Writing – original draft:** Pedro Cury, Gustavo Frigieri, Juliana Caldas.

**Writing – review & editing:** Pedro Cury, Fabio S. Taccone, Ronney B. Panerai.

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
