## [Decision Letter · Decision Letter 0]

8 Oct 2023

PONE-D-23-22748Impact of different blood pressure targets on cerebral hemodynamics in septic shock: a prospective study protocolPLOS ONE

Dear Dr. Cury,

Thank you for submitting your manuscript to PLOS ONE. After careful consideration, we feel that it has merit but does not fully meet PLOS ONE’s publication criteria as it currently stands. Therefore, we invite you to submit a revised version of the manuscript that addresses the points raised during the review process.

ACADEMIC EDITOR: All issues highlighted by expert reviewers are required and need careful revision.

We look forward to receiving your revised manuscript.

Kind regards,

Vincenzo Lionetti, M.D., PhD

Academic Editor

PLOS ONE

4. We note that Figure 2 in your submission contain copyrighted images. All PLOS content is published under the Creative Commons Attribution License (CC BY 4.0), which means that the manuscript, images, and Supporting Information files will be freely available online, and any third party is permitted to access, download, copy, distribute, and use these materials in any way, even commercially, with proper attribution. For more information, see our copyright guidelines: http://journals.plos.org/plosone/s/licenses-and-copyright.

5. We note that the original protocol that you have uploaded as a Supporting Information file contains an institutional logo. As this logo is likely copyrighted, we ask that you please remove it from this file and upload an updated version upon resubmission.

Reviewers' comments:

Reviewer's Responses to Questions

**Comments to the Author**

1. Does the manuscript provide a valid rationale for the proposed study, with clearly identified and justified research questions?

Reviewer #1: Partly

Reviewer #2: Yes

2. Is the protocol technically sound and planned in a manner that will lead to a meaningful outcome and allow testing the stated hypotheses?

Reviewer #1: Partly

Reviewer #2: Yes

3. Is the methodology feasible and described in sufficient detail to allow the work to be replicable?

Reviewer #1: No

Reviewer #2: Yes

4. Have the authors described where all data underlying the findings will be made available when the study is complete?

Reviewer #1: No

Reviewer #2: Yes

5. Is the manuscript presented in an intelligible fashion and written in standard English?

Reviewer #1: No

Reviewer #2: No

6. Review Comments to the Author

You may also provide optional suggestions and comments to authors that they might find helpful in planning their study.

Reviewer #1: Thank you for the opportunity to review the manuscript PONE-D-23-22748 by Cury et al. This protocol aims to investigate different mean arterial pressures in patients with septic shock against an expected impaired cerebral perfusion (related to autoregulatory capacity and compliance) and associations with acute organ dysfunction.

I have the following comments:

ABSTRACT:

Please explain the method rather than using the wording ‘in a program capable of evaluating the autoregulatory index (ARI), CrCP and resistance-area product (RAP).’

Avoid abbreviations and make the text as self-explanatory as possible. The Abstract should provide an adequate summary of the protocol without the need to consult the main body of text when indexed on search sites.

PROTOCOL:

Please explain why the data availability statements differ for this submission (Deidentified research data will be made publicly available when the study is completed

and published) compared to the ClinicalTrials.gov registration (PLAN TO SHARE INDIVIDUAL PARTICIPANT DATA (IPD) No) or the dissemination policy text (data kept confidential for 5 years). Please provide an explicit statement for data availability that should include the location (eg. OSF, Zenodo, Figshare or similar).

Please explain the difference in the SPRINT diagram which indicates study start in July 2023 vs. ClinicalTrials (recruitment is ongoing, 7/7/2022) and the manuscript text (commencing 1/6/2023).

Please indicate time periods for the different MAPs in the SPRINT diagram.

From the manuscript: ‘We expect that these results will present a different pattern in patients with impaired cerebral autoregulation and altered intracranial compliance, and consider whether these parameters can be modified by different MAP levels, suggesting a personalized cerebral hemodynamics approach.’ Would it not be more correct to state that different MAP can support CPP to a varying degree while the CA/ICC impairment is the same?

From the manuscript ‘The primary aim is to assess cerebral hemodynamics (ARI, CrCP, RAP

and P2/P1) in septic shock patients submitted to the 65 mmHg of MAP’. This statement needs to be elaborated. The dynamic autoregulation require a variation in MAP and this is often reduced during a MAP directed therapy. What quality criteria will be used for the dynamic autoregulation? Will the time periods be changed if there is insufficient MAP variability to assess dynamic CVAR? A time period of five minutes seems insufficient to this reviewer. Can MAP be measured from several sites (eg. radial, brachial, femoral)? The ARI is derived from a step change in MAP, how will this be assessed at the baseline MAP 65 mmHg (only once increased to 75 mmHg)? What criteria will be used to define impaired dynamic autoregulation?

Please clarify the endpoint ‘Brain dysfunction’ for the 7- and 28-day follow up.

It appears that CNS infections are not excluded (cf Table 1), is this correct? Please clarify.

The sample size is not supported by data on statistical dispersion, any minimal differences to be detected etc. No consideration of study attrition is given (for example poor TCD windows). The sample size statement as written is not adequate.

The recruitment statement does not seem conducive to consecutive enrollment but a pragmatic, logistic approach. Please explain and clarify (a pragmatic, convenience approach would be acceptable for a pilot trial).

Please elaborate on study blinding. It appears the investigators are the clinical staff involved in patient management and will also perform the subsequent analyses? Sequential numbering in such a small sample size does not guarantee blinding. Please explain and clarify (incomplete blinding would be acceptable in a pilot trial).

Please elaborate on how sedatives and analgesic use will be adjusted for. The statement ‘GCS<15… regardless of sedation’ appears counterintuitive for the study aims?

Please elaborate on procedures to adjust for adjunct therapies, importantly ventilatory support since likely to influence haemodynamic variables. While blood pressure, and hence CPP, is the primary variable of interest, how will for example differences in cardiac output, arterial CO2 and O2 be managed since they all affect autoregulation and compliance.

Please explain and support why MAP levels are not randomised but always done in the same order (if this is indeed correct). While the repeated measures ANOVA is appropriate, what post hoc testing will be performed? Will any methods be used to assess non-linear correlations? Will multivariable regressions be performed? Including adjusting for confounders / co-variates?

How will ‘missing and incorrect data’ be managed (since no imputations will be made)? This again highlights the need to explain measures in place against study attrition.

MINOR

A thorough linguistic revision for style, syntax and grammar would improve the manuscript.

Reviewer #2: I read with great interest the study protocol titled « impact of different blood pressure targets on cerebral hemodynamics in septic shock: a prospective study protocol (SEPSIS-BRAIN) ».

Sepsis-associated brain dysfunction is frequent and associated with high morbidity and mortality. The mechanisms are complex but recent studies found an association between alteration of cerebral auto regulation and cerebral dysfunction.

The authors hypothesize that cerebral auto regulation is impacted by different MAP targets, suggesting a personalized approach that could be evaluated in further large randomized studies.

They propose a one-year prospective monocentric pilot trial to address this matter. The main objective is to describe the cerebral hemodynamics profile at different pressure targets (65, 75, 85 mmHg). Secondary objective is to correlate cerebral hemodynamics with clinical severity and neurological dysfunction.

Cerebral hemodynamic is approached by two non-invasive devices: a transcranial Doppler of the middle cerebral artery, and a second innovative device evaluating intra cranial compliance via a skull micro-deformation sensor.

The study protocol addresses an important matter, with potentially major implications. It would be the first study to evaluate cerebral hemodynamics at different MAP targets in patients with septic shock. The use of an innovative device to evaluate cranial compliance in this population is an added-value.

I would recommend to improve the writing quality before publishing.

For instance, Figure 1 contains multiple errors: MAP targets during the intervention protocol are false; please add hours to “< 48 ICU admission”; cererbral artery : remove the extra R.

Some sentences must be rephrased. As an exemple: “All the patients that use vasoactive drugs will be considered to enroll, so the criteria for inclusion and exclusion will be applied (Table-1 )”

There is no major methodological issue with the study protocol. Here are listed minor concerns:

- COPD is presented as an exclusion criterion. It is a frequent condition in critically ill patient. Could you limit this exclusion criterion to COPD patients with severe hypercapnia?

- Section “Who will take informed consent?”: according to inclusion criteria, many patients could receive information and consent. The paragraph only refers to legal representatives.

- Section “Methods: data collection and management”:

o Please collect use of sedation and/or neuromuscular blocking agents since it could affect cerebral hemodynamic and brain dysfunction.

o Please collect D28 mortality. It could be interesting to analyze the association between cerebral hemodynamic and mortality.

o Evaluation of brain dysfunction could be standardized, using specific scores (CAM-ICU for instance).

7. PLOS authors have the option to publish the peer review history of their article (what does this mean?). If published, this will include your full peer review and any attached files.

Reviewer #1: No

Reviewer #2: No

---

## [Author Response · Author response to Decision Letter 0]

22 Jan 2024

Dear Vincenzo Lionetti, 

Thank you very much for conducting the review process for our manuscript PONE-D-23-22748 and for the opportunity to address the points raised by the reviewers. We found their comments relevant and these were taken into account to produce a revised version of the manuscript. We trust that you will find that our manuscript has been considerably improved, and we hope the revised version will meet your acceptance criteria. Our replies to each of the comments from the reviewers are given below where we also explain corresponding changes made to the manuscript.

Reviewer #1:

Q1. ABSTRACT: Please explain the method rather than using the wording ‘in a program capable of evaluating the autoregulatory index(ARI), CrCP and resistance-area product (RAP).

Reply: Thank you for your careful reading of the manuscript and for your suggestions. We have implemented all your recommendations. Please, see lines 37 and 41.

Q2. Avoid abbreviations and make the text as self-explanatory as possible. The Abstract should provide an adequate summary of the protocol without the need to consult the main body of text when indexed on search sites. 

Reply: We have now modified this point, thanks for your help with this.

Q3 PROTOCOL:. Please explain why the data availability statements differ for this submission (Deidentified research data will be made publicly available when the study is completed and published) compared to the ClinicalTrials.gov registration (PLAN TO SHARE INDIVIDUAL PARTICIPANT DATA(IPD) No) or the dissemination policy text (data kept confidential for 5 years). Please provide an explicit statement for data availability that should include the location (eg. OSF, Zenodo, Figshare or similar).

Reply – Many thanks for your suggestion. To address your concern, we have now modified the clinical trials.gov protocol and we will provide the data of the patients in the Zenodo. 

Q4. Please explain the difference in the SPRINT diagram which indicates study starts in July 2023 vs. ClinicalTrials(recruitment is ongoing, 7/7/2022) and the manuscript text (commencing 1/6/2023).

Reply: We encountered some issues with our equipment and found it necessary to acquire new TCD probes. Consequently, our recruitment was interrupted for a period of several months. In response to this situation, we took the necessary step of updating the information on ClinicalTrials.

Q5. Please indicate time periods for the different MAPs in the SPRINT diagram. 

R - Thank you. We have added the requested changes.

Q6. From the manuscript: ‘We expect that these results will present a different pattern in patients with impaired cerebral autoregulation and altered intracranial compliance, and consider whether these parameters can be modified by different MAP levels, suggesting a personalized cerebral hemodynamics approach.’ Would it not be more correct tostate that different MAP can support CPP to a varying degree while the CA/ICC impairment is the same?

Reply - The study aims to utilize different MAP targets (exposure) to assess CA and ICC (outcomes). We exclude subjects with neurological conditions (acute or chronic) and will describe variations in carbon dioxide, while keeping intracranial pressure constant. Assuming that other CA variables remain constant during the short data collection period, the modifying factor of CA indices would be MAP. The trial seeks to investigate whether manipulating pressure can modify CA in patients with septic shock. This is important because changes in cerebral microcirculation may occur even with normal systemic hemodynamics and could contribute to tissue hypoxia. Therefore, the first step will be to analyze the commonly targeted MAP of 65mmHg.

Q7. From the manuscript ‘The primary aim is to assess cerebral hemodynamics (ARI, CrCP, RAP and P2/P1) in septic shock patients submitted to the 65 mmHg of MAP’. This statement needs to be elaborated. The dynamic autoregulation requires a variation in MAP and this is often reduced during a MAP directed therapy. What quality criteria will be used for the dynamic autoregulation? Will the time periods be changed if there is insufficient MAP variability to assess dynamic CVAR? A time period of five minutes seems insufficient to this reviewer. Can MAP be measured from several sites (eg. radial, brachial, femoral)? The ARI is derived from a step change in MAP, how will this be assessed at the baseline MAP 65 mmHg (only once increased to 75 mmHg)? What criteria will be used to define impaired dynamic autoregulation?

Reply - This is an important point. Dynamic CA is the transient response of CBF to an acute change in BP, lasting only a few seconds (Aaslid R, Lindegaard KF, Sorteberg W, et al. Cerebral autoregulation dynamics in humans. Stroke 1989; 20: 45–52.; Tiecks FP, Lam AM, Aaslid R, et al. Comparison of static and dynamic cerebral autoregulation measurements. Stroke 1995; 26: 1014–1019.) We followed the White Paper, the CA-focused guideline and the recommendation #3 (Revised) 'Recordings of spontaneous fluctuations of arterial blood pressure and cerebral blood flow for transfer function analysis should last a minimum of 5 min, assuming stationary physiological conditions and uninterrupted good quality data' (Panerai RB, Brassard P, Burma JS, et al. Transfer function analysis of dynamic cerebral autoregulation: a CARNet white paper 2022 update. J Cereb Blood Flow Metab. 2023;43(1):3–25). The Autoregulation Index (ARI) will be the parameter adopted to express dynamic CA, with impaired dynamic CA being defined as an ARI < 4. 

Our protocol is as follows: assessment of the baseline (T0) for 5 minutes, a target of 65 mmHg for 5 minutes (T1). Next, a target of 75 mmHg for 5 minutes (T2) and a target of 85 mmHg for 5 minutes. The pressure will not remain fixed; it will be around the established targets.

In the original manuscript we did not describe the different sites for measuring invasive blood pressure. We have added this information now, thank you for your suggestion. 

Q8. Please clarify the endpoint ‘Brain dysfunction’ for the 7- and 28-day follow up.

Replay – The definition of brain dysfunction that we will employ is as follows: a Glasgow Coma Scale score of less than 15 or the presence of disorientation, disorganized thinking, or agitation as reported by the attending physician. This definition applies irrespective of sedative/analgesic use and in the absence of neurological diseases (such as dementia, cerebrovascular disease, brain tumors, or previous traumatic brain injury). We have adopted this definition, which was used by Crippa et al (Crippa, I.A., Subirà, C., Vincent, JL. et al. Impaired cerebral autoregulation is associated with brain dysfunction in patients with sepsis. Crit Care 22, 327 (2018)) and is based on the definition of sepsis-associated encephalopathy.

Data from medical records will be used to access such information. Any change in the level of consciousness or the use of antipsychotics or dexmedetomidine will be considered indicative of brain dysfunction.

The decision to choose two time points is based on the rationale that analyzing only on the 28th day or at discharge might be influenced by unmeasured confounding factors, so the seventh day may provide a more accurate assessment.

Q9. It appears that CNS infections are not excluded (cf Table 1), is this correct? Please clarify.

Reply- This is an important point. We are not including any patient with any neurological injury and this includes central nervous system infection.

Q10. The sample size is not supported by data on statistical dispersion, any minimal differences to be detected etc. No consideration of study attrition is given (for example poor TCD windows). The sample size statement as written is not adequate.

Reply – Thank you. We are estimating a 1-point difference in ARI between the different MAP targets, while maintaining the 95% confidence interval. Therefore, we need a sample of 45 patients. These data can also be seen in the following publication: Fiona G. Brodie, Emily R. Atkins, Thompson G. Robinson, Ronney B. Panerai; Reliability of dynamic cerebral autoregulation measurements using spontaneous fluctuations in blood pressure. Clin Sci (London) 1 March 2009; 116(6):513–520.

Q11. The recruitment statement does not seem conducive to consecutive enrollment but a pragmatic, logistic approach. Please explain and clarify (a pragmatic, convenience approach would be acceptable for a pilot trial).

Reply - All patients that start noradrenaline infusion at the São Rafael Hospital are transferred or are already in intensive care units. Every day, the team intensivists in charge, will report to the researcher team those patients who require new vasopressors, and the researchers will actively search the units as well. If clinical suspicion is of septic shock, the individual will enter the study's current flow, adhering to eligibility criteria. If another hypothesis is defined in the subsequent 48 hours, the patient will be excluded. Each day of the week there will be a researcher available to collect data and interview the patient or their family. Individuals can be recruited within 48 hours of the initiation of vasopressors, making data collection feasible. 

Q12. Please elaborate on study blinding. It appears the investigators are the clinical staff involved in patient managementand will also perform the subsequent analyses? Sequential numbering in such a small sample size does not guarantee blinding. Please explain and clarify (incomplete blinding would be acceptable in a pilot trial).

Reply: Two researchers are involved in the data collection. During data collection, they have access to Transcranial Doppler parameters, such as mean velocity and pulsatility index. However, ARI from Transfer Function Analysis, CrCP, and compliance parameters for non-invasive ICP are needed for further analysis. 

The pilot study is not blind. at the time of data collection, it will be impossible to know the value of the ARI or CrCP. Moreover, during data analysis, when runnint the software, the classification of the patient could be blind to the operator.

Q13. Please elaborate on how sedatives and analgesic use will be adjusted for. The statement ‘GCS<15… regardless of sedation’ appears counterintuitive for the study aims?

Reply: This term ‘GCS<15… regardless of sedation’ was used for brain dysfunction (BD). Independent, will be describe like BD if GCS < 15 (in the absence of neurologic disorders) or if disorientation, disorganized thinking or agitation were reported by the attending physician, regardless of sedative/analgesic. We don’t pretend to be adjusted for this variable because if the patient exhibits agitation despite the use of sedatives and analgesics, the data will be recorded in the same way as for those who were not using them. 

The primary objective is to evaluate cerebral dysfunction associated with sepsis without making a distinction between sedated and non-sedated individuals. Nonetheless, a secondary analysis can be conducted to ascertain if sedated patients present a similar profile of brain dysfunction.

Q14 Please elaborate on procedures to adjust for adjunct therapies, importantly ventilatory support since likely to influence haemodynamic variables. While blood pressure, and hence CPP, is the primary variable of interest, how will for example differences in cardiac output, arterial CO2 and O2 be managed since they all affect autoregulation and compliance.

Reply: Thank you for raising this very relevant point. As we are aware of the significant influence of CO2 and O2 levels on cerebral hemodynamics, all patients are required to have an arterial catheter. Blood gas analysis will be conducted both before and after the protocol to observe the variability of gases in each patient. For mechanically ventilated individuals, capnography will be employed, and the value will be recorded at the beginning of each blood pressure target.

Regarding cardiac dysfunction, we will describe the presence or absence of cardiac dysfunction, dichotomized based on the left ventricular ejection fraction. We will utilize the patient's most recent echocardiogram for this purpose. (performed during the same hospitalization or prior). We will also describe the number of patients with severe diastolic dysfunction.

Q15. Please explain and support why MAP levels are not randomised but always done in the same order (if this is indeed correct). While the repeated measures ANOVA is appropriate, what post hoc testing will be performed? Will any methods be used to assess non-linear correlations? Will multivariable regressions be performed? Including adjusting for confounders / co-variates?

Reply- We initially did not anticipate that randomizing the MAP levels would alter the autoregulation indices. Our rationale was based on the distinct time intervals required to attain individual blood pressure targets, thus assuming that preceding blood pressure levels would not exert influence on the assessed target.

 Changes in cerebral hemodynamic (ie: ARI/ CrCP, P1P2 ) at T1, T2 and T3 will be assessed with repeated measures ANOVA. 

Furthermore, logistic regression with both categorical and continuous independent variables will be used to build predictive models for the occurrence of impaired cerebral autoregulation and brain dysfunction. The binary classification (presence + vs. absent) will be used as the outcome variable. Logistic regression models for the presence or absence of impaired cerebral autoregulation and brain dysfunction will be constructed using those variables with statistically significant differences between groups. 

Q16. How will ‘missing and incorrect data’ be managed (since no imputations will be made)? This again highlights the need to explain measures in place against study attrition.

Reply: Thank you. We will modify and impute the missing data by the sample mean.

Q17 MINOR. A thorough linguistic revision for style, syntax and grammar would improve the manuscript.

Reply - Thank you, we will do a linguistic review. 

Reviewer #2

Q18. I would recommend to improve the writing quality before publishing. For instance, Figure 1 contains multiple errors: MAP targets during the intervention protocol are false; please addhours to “< 48 ICU admission”; cererbral artery : remove the extra R.

Reply: Thank you for your careful reading of the manuscript. We have corrected these mistakes.

Q.19. Some sentences must be rephrased. As an exemple: “All the patients that use vasoactive drugs will be considered to enroll, so the criteria for inclusion and exclusion will be applied (Table-1 )”

Reply: We have corrected this table legend. 

Q.20 There is no major methodological issue with the study protocol. Here are listed minor concerns:- COPD is presented as an exclusion criterion. It is a frequent condition in critically ill patient. Could you limit this exclusion criterion to COPD patients with severe hypercapnia?

Reply: We agree. The understanding that carbon dioxide (CO2), as the principal metabolic factor, plays a substantial role in influencing cerebral hemodynamics is well-established. Hypercapnia, in particular, has the potential to adversely impact cerebral autoregulation (The effect of hypercapnia on the directional sensitivity of dynamic cerebral autoregulation and the influence of age and sex. Panerai RB, J Cereb Blood Flow Metab, 2023). We will exclude COPD with hypercapnia. 

Q21. Section “Who will take informed consent?”: according to inclusion criteria, many patients could receive information and consent. The paragraph only refers to legal representatives.

Reply: Informed consent will be obtained from the patient or their legal representative. Please, see line 162. 

Q22. Section “Methods: data collection and management Please collect use of sedation and/or neuromuscular blocking agents since it could affect cerebral hemodynamicand brain dysfunction.

Reply: Details will be provided regarding the administration of sedation, the specific drug utilized, and the use of neuromuscular blocking agents.

Q23. Please collect D28 mortality. It could be interesting to analyze the association between cerebral hemodynamic and mortality.

Reply: Thank you very much for this valuable insigh

---

## [Decision Letter · Decision Letter 1]

2 Feb 2024

PONE-D-23-22748R1Impact of different blood pressure targets on cerebral hemodynamics in septic shock: a prospective study protocol - SEPSIS-BRAINPLOS ONE

Dear Dr. Cury,

Thank you for submitting your manuscript to PLOS ONE. After careful consideration, we feel that it has merit but does not fully meet PLOS ONE’s publication criteria as it currently stands. Therefore, we invite you to submit a revised version of the manuscript that addresses the points raised during the review process.

**ACADEMIC EDITOR: **All issues raised by expert reviewer are required.

We look forward to receiving your revised manuscript.

Kind regards,

Vincenzo Lionetti, M.D., PhD

Academic Editor

PLOS ONE

Reviewers' comments:

Reviewer's Responses to Questions

**Comments to the Author**

1. Does the manuscript provide a valid rationale for the proposed study, with clearly identified and justified research questions?

Reviewer #3: Yes

2. Is the protocol technically sound and planned in a manner that will lead to a meaningful outcome and allow testing the stated hypotheses?

Reviewer #3: No

3. Is the methodology feasible and described in sufficient detail to allow the work to be replicable?

Reviewer #3: No

4. Have the authors described where all data underlying the findings will be made available when the study is complete?

Reviewer #3: Yes

5. Is the manuscript presented in an intelligible fashion and written in standard English?

Reviewer #3: No

6. Review Comments to the Author

You may also provide optional suggestions and comments to authors that they might find helpful in planning their study.

Reviewer #3: Dear Editor,

Thank you for the opportunity to provide a review of Manuscript PONE-D-23-22748 entitled "Impact of different blood pressure targets on cerebral hemodynamics in septic shock: a prospective study protocol - SEPSIS-BRAIN". My comments relate primarily to the adequacy of the implementation and reporting of epidemiologic and statistical procedures.

The quality of the technical English was appropriate and offered no bar to my evaluation of the manuscript. Nevertheless, there are many instances of spelling, grammatical, and syntactical errors—too many for me to identify. The authors need to conduct a thorough round of copyediting to identify and correct these mistakes.

Overall, the statistical approaches described in this manuscript are poorly described. They are listed generically. This lack of specificity means that the methods cannot be replicated and, hence, cannot be assessed as appropriate.

# Define the primary outcome or outcomes clearly and with specificity.

First, the authors must be clear when they specify the primary outcome. As it currently stands, the authors state that the primary outcome is "changes in cerebral haemodynamic variables and other parameters". This is non-specific. This is not clarified in other areas in the manuscript. The sample size is calculated against ARI, so I surmise that this is one of the primary outcomes. From Figure 1, I further surmise that ARI is one of three measurements from TCD, the other two being CrCP and RAP. Only one measurement of interest is taken from B4C, which is P2/P1. At no time are these four measurements, taken together, identified as the "cerebral hemodynamics" to which the authors refer. This is inadequate. The authors need to be clear about which of the measurements being taken are used as the primary outcome or outcomes.

Once identified, these measurements must be declared clearly in the statement about the study outcome. Thus, instead of the generic "The primary endpoint is to assess the impact of a target MAP of 65 mmHg on cerebral hemodynamics in septic shock patients", the authors should state "The primary endpoint is to assess the impact of a target MAP of 65 mmHg on ARI, CrCP, RAP via TCD and P2/P1 via B4C in septic shock patients", if this is indeed the case. As I said, the manuscript wasn't clear, although the abstract identifies just the case.

# The intervention is unclear

The authors state that the interventions being applied are three different MAP targets, yet they do not describe how these targets will be achieved. This is a major omission.

# The authors do not describe the order in which the targets are achieved

I assume that the target MAP levels will be achieved in sequence after baseline -- 65, then 75, then 85. This is not stated.

If this is the case, how will the authors account for the effect of the baseline MAP and the first MAP target of 65? Consider Patient A with a baseline MAP of 87. The authors will REDUCE the patient's MAP from 87 to 65 and hold it steady for five minutes to take readings.

Consider Patient B with a baseline MAP of 35. In this setting, the authors will INCREASE the patient's MAP to 65 and hold it steady for five minutes for readings.

Consider Patient C with a baseline MAP of 65. In this setting, the authors will simply repeat the readings from baseline.

Do the authors assume that these three patients are qualitatively similar in their responses? If not, then this situation must be considered in the analysis.

In the case of Patient A, the authors intend to decrease the MAP from 87 to 65. Why not stop at 85, then 75, then 65 and take readings at each stop?

It is also not clear what happens in between readings. Take Patient B's progress. The patient's MAP will be raised to 65 from 35. Will the MAP be held steady until it is time to go to 75 or will be patient's MAP be allowed to reduce to baseline (effectively a washout period of sorts) before moving to the 75 target?

# The sample size statements are unreplicable.

A previous reviewer noted that the original manuscript's description of the sample size calculation was inadequate. I'm sorry to say that the authors' response is inadequate. The calculation of the sample size is important and the revisions do not allow for confirmation of the size estimates.

The authors cite a separate article. This is inappropriate. The current manuscript must stand alone. The authors must identify which pieces of information they took from the reference and show how they arrived at the current sample size. In particular, they must justify the following:

(1) that a 1-point increase in ARI is a clinically relevant change, and

(2) the variability (expressed as a standard deviation or variance) around the ARI readings at the target MAP of 65.

# The authors have not adequately addressed the issue of confounding.

The authors' response to a previous reviewer on control for confounding variables is inadequate. It is likely that the sample size estimate, as vague as it is, will not allow for the adjustment of many variables. The authors must pre-specify the variables that they expect to use in the ANOVA and regression models that they will build. They must use their clinical knowledge and expertise to identify these. Previous studies might provide some clues about what characteristics will be relevant. For example, do the authors expect that males and females will have readings that are different? What about body weight or the presence of specific vasopressors in the management of sepsis? All of these must be clearly defined BEFORE the experiment is conducted.

# Linear regression may be inappropriate

The authors state that "Linear regression can be used to formulate a model based on different pressure targets." This may or may not be appropriate depending on the way that the model is operationalised. At present there is not enough information to tell whether this is appropriate because the statement is inadequately specified.

The authors must use a regression model that accounts for the non-independence of measurements. That is, each patient serves as his or her control. The authors cannot simply ignore this major feature of their data since independence of measurements is one of the assumptions of linear regression and it is voided by the authors' study design. The authors must be clear about how they plan to account for this.

The authors may need to seek the assistance of a professional biostatistician to refine their methods.

# Recommendation

I cannot support the acceptance of this manuscript for publication in the Journal until these issues are considered and addressed.

Thank you.

7. PLOS authors have the option to publish the peer review history of their article (what does this mean?). If published, this will include your full peer review and any attached files.

Reviewer #3: No

---

## [Author Response · Author response to Decision Letter 1]

1 Apr 2024

Dear Dr. Vincenzo Lionetti, 

Thank you very much for conducting the review process for our manuscript PONE-D-23-22748R1 and for the opportunity to address the points raised by the reviewers. We found their comments relevant and these were taken into account to produce a revised version of the manuscript. We trust that you will find that our manuscript has been considerably improved, and we hope the revised version will meet your acceptance criteria. Our replies to each of the comments from the reviewers are given below where we also explain corresponding changes made to the manuscript.

Comments to the Author:

Q2. Is the protocol technically sound and planned in a manner that will lead to a meaningful outcome and allow testing the stated hypotheses?

Comment: The manuscript should describe the methods in sufficient detail to prevent undisclosed flexibility in the experimental procedure or analysis pipeline, including sufficient outcome-neutral conditions (e.g. necessary controls, absence of floor or ceiling effects) to test the proposed hypotheses and a statistical power analysis where applicable. As there may be aspects of the methodology and analysis which can only be refined once the work is undertaken, authors should outline potential assumptions and explicitly describe what aspects of the proposed analyses, if any, are exploratory. 

Replay: Thank you for the clarifications. We have made some adjustments and highlighted key points for clarity. Our protocol is a pilot project aimed at evaluating physiological data. In this context, we realized the need to refine our primary objective.

Our primary outcome is to characterize cerebral hemodynamics in patients with septic shock across different blood pressure targets. This characterization will be conducted using Transcranial Doppler and B4C. The ARI, CrCp and RAP will be assessed by transcranial Doppler RAP, while the P2/P1 ratio by B4C device.

Q3. Is the methodology feasible and described in sufficient detail to allow the work to be replicable?

Comment: Descriptions of methods and materials in the protocol should be reported in sufficient detail for another researcher to reproduce all experiments and analyses. The protocol should describe the appropriate controls, sample size calculations, and replication needed to ensure that the data are robust and reproducible.

Reply: Thank you for your thoroughness in describing the methodology. We have provided detailed information on patient selection, sample collection location, duration, intervention procedures, and data collection. Additionally, we have included extensive details on measurements of cerebral hemodynamics, which have been well-documented in the literature, including critical care patients (Panerai, RB J, Cereb Blood Flow Metab, 2023; Caldas J, Annals of Intensive Care Medicine 2019; Caldas J, Shock, 2019; Bergman L, Am J Obstet Gynecol. 2022; Salinet AS, J Cereb Blood Flow Metab. 2019; Simpson DM, J Cereb Blood Flow Metab. 2022; Caldas J, Neurophysiology, 2019;) .Additionally, we have included details to enhance reproducibility, such as the sequence for achieving mean arterial pressure targets. We will not include control groups in our study, similar to the approach taken by others researchers (Crippa et al. Critical Care 2018; Pierrakos et al. Annals of Intensive Care 2013,). Our study is meticulously crafted to elucidate physiological parameters within a pilot investigation. Its principal objective is to characterize the pattern of cerebral hemodynamics within this specific population. This pilot initiative aims to provide foundational data for future studies, promoting adherence to the same protocol and facilitating the utilization of amassed data for sample size calculations.

Q5 Is the manuscript presented in an intelligible fashion and written in standard English?

Reply – Thank you, we have corrected English errors.

Review Comments to the Author

# Define the primary outcome or outcomes clearly and with specificity.

Comment: First, the authors must be clear when they specify the primary outcome. As it currently stands, the authors state that the primary outcome is "changes in cerebral hemodynamic variables and other parameters''. This is non-specific.This is not clarified in other areas in the manuscript. The sample size is calculated against ARI, so I surmise that this is one of the primary outcomes. From Figure 1, I further surmise that ARI is one of three measurements from TCD, the other two being CrCP and RAP. Only one measurement of interest is taken from B4C, which is P2/P1. At no time are these four measurements, taken together, identified as the "cerebral hemodynamics'' to which the authors refer. This is inadequate. The authors need to be clear about which of the measurements being taken are used as the primary outcome or outcomes.

Replay: Thank you for the feedback; I'm glad to hear that the comments were helpful in refining our project. We have made adjustments accordingly.

Our primary objective now focuses on evaluating cerebral hemodynamics across different blood pressure targets. It's important to note that all components of cerebral hemodynamics will constitute the primary outcome. This study serves as a pilot project, and the data obtained will be instrumental in guiding future research focusing on specific indices as primary objectives.

Additionally, we have included the 65 mmHg blood pressure target as a secondary objective in our study. This is in line with current guidelines and will contribute to the overall description of cerebral hemodynamics.

# The intervention is unclear

Comment: The authors state that the interventions being applied are three different MAP targets, yet they do not describe how these targets will be achieved. This is a major omission.

.Replay: Thank you for bringing attention to this point. I was thinking of describing it as follows: The primary blood pressure target will be the mean arterial pressure (MAP) level of the patient at the time of collection, with subsequent targets invariably following the order of 65 > 75 > 85 to ensure consistency in the collection pattern.

# The sample size statements are irreplaceable.

Comment: A previous reviewer noted that the original manuscript's description of the sample size calculation was inadequate. I'm sorry to say that the authors' response is inadequate. The calculation of the sample size is important and the revisions do not allow for confirmation of the size estimates.

The authors cite a separate article. This is inappropriate. The current manuscript must stand alone. The authors must identify which pieces of information they took from the reference and show how they arrived at the current sample size. In particular, they must justify the following:

(1) that a 1-point increase in ARI is a clinically relevant change, and

(2) the variability (expressed as a standard deviation or variance) around the ARI readings at the target MAP of 65.

Replay: Thank you for your comment. We cited the article separately to highlight the observed differences in pressure between groups as an example. Our study is indeed a pilot project, with a sample size we deemed feasible to collect within this timeframe. The physiological data obtained from this study are intended to serve as a foundation for subsequent research endeavors. Our aim is to facilitate the development of larger studies that can follow this protocol and derive a correct sample size based on the generated data. 

# Linear regression may be inappropriate

Comment: The authors state that "Linear regression can be used to formulate a model based on different pressure targets." This may or may not be appropriate depending on the way that the model is operationalised. At present there is not enough information to tell whether this is appropriate because the statement is inadequately specified.

The authors must use a regression model that accounts for the non-independence of measurements. That is, each patient serves as his or her control. The authors cannot simply ignore this major feature of their data since independence of measurements is one of the assumptions of linear regression and it is voided by the authors' study design. The authors must be clear about how they plan to account for this.

The authors may need to seek the assistance of a professional biostatistician to refine their methods.

Replay:I appreciate your input. Our intention is to be as transparent as possible.Changes in cerebral haemodynamics indexes at different times, T0, T1, T2 and T3 will be assessed with repeated measures ANOVA. Additionally, in the chapter that deals with the statistical model, we will add the fact that the measurements will be taken repeatedly and that there will be interdependence, and therefore, the most suitable model is mixed-effects, nonlinear regression models will be tested.

---

## [Decision Letter · Decision Letter 2]

11 Apr 2024

PONE-D-23-22748R2Impact of different blood pressure targets on cerebral hemodynamics in septic shock: a prospective pilot study protocol - SEPSIS-BRAINPLOS ONE

Dear Dr. Cury,

Thank you for submitting your manuscript to PLOS ONE. After careful consideration, we feel that it has merit but does not fully meet PLOS ONE’s publication criteria as it currently stands. Therefore, we invite you to submit a revised version of the manuscript that addresses the points raised during the review process.

**ACADEMIC EDITOR: all issues from reviewer are required>==============================**

**Please submit your revised manuscript by May 26 2024 11:59PM. If you will need more time than this to complete your revisions, please reply to this message or contact the journal office at plosone@plos.org. When you're ready to submit your revision, log on to https://www.editorialmanager.com/pone/ and select the 'Submissions Needing Revision' folder to locate your manuscript file**.

**Please include the following items when submitting your revised manuscript:****A rebuttal letter that responds to each point raised by the academic editor and reviewer(s). You should upload this letter as a separate file labeled 'Response to Reviewers'.****A marked-up copy of your manuscript that highlights changes made to the original version. You should upload this as a separate file labeled 'Revised Manuscript with Track Changes'.****An unmarked version of your revised paper without tracked changes. You should upload this as a separate file labeled 'Manuscript'.******If applicable, we recommend that you deposit your laboratory protocols in protocols.io to enhance the reproducibility of your results. Protocols.io assigns your protocol its own identifier (DOI) so that it can be cited independently in the future. For instructions see: https://journals.plos.org/plosone/s/submission-guidelines#loc-laboratory-protocols. Additionally, PLOS ONE offers an option for publishing peer-reviewed Lab Protocol articles, which describe protocols hosted on protocols.io. Read more information on sharing protocols at https://plos.org/protocols?utm_medium=editorial-email&utm_source=authorletters&utm_campaign=protocols**.

**We look forward to receiving your revised manuscript.**

**Kind regards,**

**Vincenzo Lionetti, M.D., PhD**

**Academic Editor**

**PLOS ONE**

**Journal Requirements:**

**Please review your reference list to ensure that it is complete and correct. If you have cited papers that have been retracted, please include the rationale for doing so in the manuscript text, or remove these references and replace them with relevant current references. Any changes to the reference list should be mentioned in the rebuttal letter that accompanies your revised manuscript. If you need to cite a retracted article, indicate the article’s retracted status in the References list and also include a citation and full reference for the retraction notice**.

****

**Reviewers' comments**:

**Reviewer's Responses to Questions**

**Comments to the Author**

**1. Does the manuscript provide a valid rationale for the proposed study, with clearly identified and justified research questions?**

**The research question outlined is expected to address a valid academic problem or topic and contribute to the base of knowledge in the field.**

**Reviewer #3: Yes**

**2. Is the protocol technically sound and planned in a manner that will lead to a meaningful outcome and allow testing the stated hypotheses?**

**The manuscript should describe the methods in sufficient detail to prevent undisclosed flexibility in the experimental procedure or analysis pipeline, including sufficient outcome-neutral conditions (e.g. necessary controls, absence of floor or ceiling effects) to test the proposed hypotheses and a statistical power analysis where applicable. As there may be aspects of the methodology and analysis which can only be refined once the work is undertaken, authors should outline potential assumptions and explicitly describe what aspects of the proposed analyses, if any, are exploratory.**

**Reviewer #3: Yes**

**3. Is the methodology feasible and described in sufficient detail to allow the work to be replicable?**

**Descriptions of methods and materials in the protocol should be reported in sufficient detail for another researcher to reproduce all experiments and analyses. The protocol should describe the appropriate controls, sample size calculations, and replication needed to ensure that the data are robust and reproducible.**

**Reviewer #3: Yes**

**4. Have the authors described where all data underlying the findings will be made available when the study is complete?**

**The PLOS Data policy requires authors to make all data underlying the findings described in their manuscript fully available without restriction, with rare exception, at the time of publication. The data should be provided as part of the manuscript or its supporting information, or deposited to a public repository. For example, in addition to summary statistics, the data points behind means, medians and variance measures should be available. If there are restrictions on publicly sharing data—e.g. participant privacy or use of data from a third party—those must be specified.**

**Reviewer #3: Yes**

**5. Is the manuscript presented in an intelligible fashion and written in standard English?**

**PLOS ONE does not copyedit accepted manuscripts, so the language in submitted articles must be clear, correct, and unambiguous. Any typographical or grammatical errors should be corrected at revision, so please note any specific errors here.**

**Reviewer #3: No**

**6. Review Comments to the Author**

**Please use the space provided to explain your answers to the questions above and, if applicable, provide comments about issues authors must address before this protocol can be accepted for publication. You may also include additional comments for the author, including concerns about research or publication ethics**.

**You may also provide optional suggestions and comments to authors that they might find helpful in planning their study**.

**(Please upload your review as an attachment if it exceeds 20,000 characters)**

**Reviewer #3: Please conduct a thorough copyediting pass to correct the numerous grammatical and spelling errors in the manuscript.**

**7. PLOS authors have the option to publish the peer review history of their article (what does this mean?). If published, this will include your full peer review and any attached files**.

**If you choose “no”, your identity will remain anonymous but your review may still be made public**.

**Do you want your identity to be public for this peer review? For information about this choice, including consent withdrawal, please see our Privacy Policy.**

**Reviewer #3: No**

****

**While revising your submission, please upload your figure files to the Preflight Analysis and Conversion Engine (PACE) digital diagnostic tool, https://pacev2.apexcovantage.com/. PACE helps ensure that figures meet PLOS requirements. To use PACE, you must first register as a user. Registration is free. Then, login and navigate to the UPLOAD tab, where you will find detailed instructions on how to use the tool. If you encounter any issues or have any questions when using PACE, please email PLOS at figures@plos.org. Please note that Supporting Information files do not need this step.**

---

## [Author Response · Author response to Decision Letter 2]

9 May 2024

Dear Vincenzo Lionetti, 

Thank you very much for conducting the review process for our manuscript PONE-D-23-22748R1 and for the opportunity to address the points raised by the reviewers. We found their comments relevant and these were taken into account to produce a revised version of the manuscript. We trust that you will find that our manuscript has been considerably improved, and we hope the revised version will meet your acceptance criteria. Our replies to each of the comments from the reviewers are given below where we also explain corresponding changes made to the manuscript.

Comments to the Author:

Q5 Is the manuscript presented in an intelligible fashion and written in standard English?

Reply – Thank you, we've done an extensive review, we hope it's satisfactory.

---

## [Editor Report · Decision Letter 3]

13 May 2024

Impact of different blood pressure targets on cerebral hemodynamics in septic shock: a prospective pilot study protocol - SEPSIS-BRAIN

PONE-D-23-22748R3

Dear Dr. Cury,

We’re pleased to inform you that your manuscript has been judged scientifically suitable for publication and will be formally accepted for publication once it meets all outstanding technical requirements.

Kind regards,

Vincenzo Lionetti, M.D., PhD

Academic Editor

PLOS ONE
---

## [Editor Report · Acceptance letter]

13 Jun 2024

PONE-D-23-22748R3 

PLOS ONE

Dear Dr. Cury, 

I'm pleased to inform you that your manuscript has been deemed suitable for publication in PLOS ONE. Congratulations! Your manuscript is now being handed over to our production team.

Kind regards, 

on behalf of

Prof. Vincenzo Lionetti 

Academic Editor

PLOS ONE